

# 1 Statistical characteristics of non-volcanic tremor distributions along the Mexican
# 2 Subduction Zone

**Quetzalcoatl Rodríguez-Pérez[1,2], Víctor H. Márquez-Ramírez[2], F. Ramón Zúñiga[2]**
[1] Secretaría de Ciencia, Humanidades, Tecnología e Innovación, Mexico City, Mexico.
[2] Instituto de Geociencias, Universidad Nacional Autónoma de México, Juriquilla, Querétaro, Mexico.
**Correspondence:** Quetzalcoatl Rodríguez-Pérez (quetza@geociencias.unam.mx)
**Abstract.**
We analyze statistical characteristics of non-volcanic tremor sequences determined in the Mexican
subduction zone. To achieve this objective, we used different techniques such as the Gutenberg-Richter
relationship, non-extensive statistics, and the multifractal detrended moving average analysis to extract
information on the magnitude and interevent-time distributions. The $b$-value results reveal that $b$
fluctuates from 1.25 to 2.42, with the highest values corresponding to the plate interface down-dip
regions. On the other hand, the $q$-value shows an inverse behavior, having the highest values in the
inter-plate coupling region. Similar to tectonic earthquakes, the non-volcanic tremor sequences show a
multifractal structure in magnitude and interevent data. The multifractality analysis suggests that
multifractality may be associated with long-term correlations, the probability distribution of the data,
and the presence of nonlinearities. Regarding the existence of apparent and intrinsic multifractality, our
results indicate that both sources are present in the sequences, with the former being the most common.
Our estimates of the Hurst exponent are in the interval of 0.65 to 1.06, with the majority indicating an
exceptionally high persistent memory ($H > 0.95$). With respect to the estimation of the distribution that
better describes the interevent sequences, we find that most sequences can be described by a
Lognormal distribution and, to a lesser extent, by a Gamma distribution. Our investigation also showed
that observations of the duration of tectonic tremors present a large scatter, resulting in scaling





relationships with low values of the determination coefficient. The source of this variability may be
related to the generation mechanism of NVT or to the process of detection and description of the
signals.
**Keywords** Non-volcanic tremor, *b*-value, non-extensive statistics, multifractality, Mexico
**1 Introduction**
Non-volcanic tremor (NVT) is an enigmatic seismic phenomenon that has become a relevant topic of
study, mainly in subduction zones. NVTs are emergent seismic signals of long duration and low
amplitude (Obara, 2002). Previous studies have shown that NVTs are composed of repeating small
low-frequency earthquakes, commonly accompanied by very-low-frequency earthquakes that, in both
cases, involve shear failure and slip (Shelly et al., 2007; Bostock et al., 2015; Gomberg et al., 2016a, b).
NVTs mainly occur within the plate interface close to the shallow and deep edges locked regions. This
phenomenon can also be observed in other tectonic environments, such as in convergent regions (e.g.,
Taiwan, Tang et al., 2010) or continental transform zones (e.g., California, Nadeau and Dolenc, 2005;
Shelly, 2010). Fluids are implicated in generating NVTs in a similar form to regular volcanic tremors
(Obara, 2002). Pressure and temperature conditions at subduction zones, in addition to dehydration,
control the occurrence of NVT (Yoshioka et al., 2008; Peacock, 2009). It has been reported that NVTs
occur in regions of high pore-fluid pressure (Shelly et al., 2006). Other factors are important in
generating tectonic tremors, such as the material composition in the overriding plate above the tremor
zone in the wedge mantle and the metamorphism of subducting seamounts (Wada et al., 2008). Non-
volcanic tremors commonly exhibit episodic activity, with periods of intense activity lasting from days
to weeks, interrupted by periods of scarce activity that last for months, during which there are few or no
tremor episodes (Obara, 2002). Regularly, NVTs coexist with slow earthquakes; both phenomena share



temporal and spatial occurrence (Rogers and Dragert, 2003). Another important feature of NVTs is that
they can be triggered by large earthquakes, such as the case of the 2002 Denali earthquake ($M_W$ = 7.8)
in Alaska (Rubinstein et al., 2007) and the 2003 Takachi-oki earthquake ($M_W$ = 8.1) in Japan
(Miyazawa and Mori, 2005), in both cases these earthquakes, activated NVT activity.
Tectonic tremors have been reported along the Middle America Trench in Mexico in Guerrero (Payero
et al., 2008; Cruz-Atienza et al., 2015; Villafuerte and Cruz-Atienza, 2017; Husker et al., 2019; Plata-
Martínez et al., 2021; Chen et al., 2025), Jalisco-Colima (Ide, 2012), and Oaxaca (Brudzinski et al.,
2010; Husker et al., 2019) states. In Guerrero, NVTs have been detected at about 200 km downdip, at
depths of 40-50 km, and within a few kilometers from the trench, at depths of 10-16 km (Plata-
Martínez et al., 2021; Chen et al., 2025). Similar to the Guerrero case, in Oaxaca, NVTs are located
approximately 150-200 km from the trench at depths of 30-50 km (Brudzinski et al., 2010). On the
other hand, in Western Mexico, NVTs occur in a narrow band oriented parallel to the trench, mostly at
depths of 30-50 km (Ide, 2012). Much of the effort in studying tectonic tremors is directed toward
locating events, generating catalogs, and their tectonic interpretation. Few studies have focused on
determining the statistical characteristics of event occurrence. For example, Kao et al. (2010)
determined the $b$-value of NVT sequences in northern Cascadia ($M_W$ ~1.0 - 1.7), finding that the $b$-
value appears to be 1. Staudenmaier et al. (2019) estimated the $b$-value of NVTs in the San Andreas
Fault (0.44 < $M_W$ < 2.7), obtaining a $b$-value of 2.52. In contrast, other studies reported extremely high
$b$-values ($b$ > 5 with 1.5 < $M_W$ < 2.7; $b$ = 4.2 with 0.3 < $M_W$ < 1.5) (Sweet et al., 2014; Bostock et al.,
2015). In this article, we studied the statistical features of NVT sequences generated at the Mexican
subduction zone. We analyzed the Gutenberg-Richter relationship, non-extensive statistics, and
multifractality of magnitude and interevent-time distributions. Our results characterize particular
statistical information of NVTs associated to subduction zones which may help shade light on their



generation.
**2 Tectonic setting**
The Mexican subduction zone (MSZ) is located along the border of three tectonic plates: the Cocos
(CO), North American (NA), and Rivera (RI) plates, respectively (Fig. 1). Convergence rates predicted
from the NUVEL1-A model (DeMets et al., 1994) fluctuate from 2.0 to 6.8 cm/yr for the RI-NA
convergence in the north (Jalisco-Colima) to the CO-NA convergence in the south (Oaxaca),
respectively (Fig. 1). The rupture areas of the previous largest earthquakes exhibit a seismic gap in the
MSZ, known as the Guerrero seismic gap (GG). The GG is a 200 km long segment in the CO-NA plate
boundary (Fig. 1). The gap is capable of producing an earthquake with a magnitude of 8.1-8.4 if the
entire gap were ruptured in a single event (Singh and Mortera, 1991). The gap has not experienced a
significant event ($M > 7$) since 1911. The geometry of the subducted slab varies from north to south. In
the Jalisco region, the RI plate subducts at a steep angle (~ 50º), and then the dip angle of the CO plate
decreases gradually toward the southeast. In the Gurrero-Oaxaca region, the subducted slab is almost
subhorizontal (Pardo and Suárez, 1995; Pérez-Campos et al., 2008). As mentioned above, non-volcanic
tremors occur at certain regions within the subduction zone regime. They appear to signal the transition
from creep to locking trench parallel segments (Chen et al., 2025).
**3 Data and methods**
**3.1 Data**
We retrieved non-volcanic tremor catalogs for events from six different sequences along the Mexican
subduction zone, as reported in previous studies (Ide, 2012; Idehara et al., 2014; Husker et al., 2019;
Plata-Martínez et al., 2021; Chen et al., 2025). We studied NVT events from 2005 to 2019, with
hypocentral depths lower than 50 km and magnitudes ranging from -0.8 to 3.65 (Table 1). Sequences 1,





3 and 4 took place down dip from the coupling plate interface of the Guerrero segment. Sequences 5
and 6 occurred at the coupling plate interface of that segment. Sequence 2 was located further west, at
the boundary between the subduction regimes of Rivera and Cocos plates (Fig. 2). The information
reported in these catalogs is the following: origin time, hypocentral location, moment magnitude ($M_\mathrm{W}$),
and time duration for sequences 1 to 3 and 5. In the case of sequence 4, locations were performed by
single local seismic stations, resulting in only information about origin time and hypocentral location.
Similar to the previous case, in sequence 6, event magnitudes are not reported. Although magnitudes
are not available for sequences 4 and 6, these catalogs contain sufficient information to study the
interevent time. Hypocenters of NVTs were located in the States of Guerrero and Jalisco and were
detected mainly through temporal seismic networks (Sequences 1, 2, 3, 5, and 6).
**3.2 Methods**
**3.2.1 Estimation of the *b*-value**
The earthquake frequency-magnitude distribution (FMD) is commonly described by the Gutenberg-
Richter law (Gutenberg and Richter, 1944):
$$\log_{10} N(M) = a + bM \quad , \tag{1}$$
where $N$ is the number of earthquakes $\geq M$ above the magnitude of completeness ($M_\mathrm{c}$), $a$ and $b$ are
constants that describe the earthquake productivity and the proportion of small to large events,
respectively. Globally, the *b*-value is about one on average (Lay and Wallace, 1995). Fluctuations from
this value are due to several factors, such as fluid pressure (Henderson et al., 1994), heterogeneity in
the fault zone (Mogi, 1963), thermal gradient (Warren and Latham, 1970), variations in the state of
stress (Schorlemmer et al., 2005; Scholz, 2015). The *b*-value is determined using the maximum





likelihood method proposed by Aki (1965). The equation that describes this estimator is the following
$$b = \frac{\log_{10} e}{\bar{M} - \left( M_c - \Delta M / 2 \right)} \quad ,$$  (2)
where $M_c$ is the catalog completeness magnitude, $\bar{M}$ is the average magnitude with a magnitude
greater than $M_c$, and ΔM is the magnitude binning interval. We determined $b$-values for NVT sequences
with reported magnitudes (sequences 1 to 3, and 5), the results are shown in Fig. 3 and Table 2.
**3.2.2 Non-extensive statistical analysis**
Non-extensive statistical mechanics (NESM) is a theoretical framework for analyzing non-equilibrium
complex systems, such as earthquake phenomena. In the case of systems showing long-range
correlations, memory, or fractal properties, NESM becomes the most suitable mathematical framework
(Tsallis, 2009). In that context, Sotolongo-Costa and Posadas (2004) introduced the fragment asperity
model. In this model, the release of seismic energy (ε) is associated with the size of $r$ fragments filling
the space between irregular fault interfaces. Silva et al. (2006) used a volumetric relationship between
seismic energy and fragment size in the form of $\varepsilon \propto r^3$ , under this assumption, the cumulative
distribution of the number of earthquakes $N$ with a magnitude greater than $M$ is
$$\log \frac{N > M}{N} = \frac{(2-q)}{(1-q)} \log \left[ 1 - \left( \frac{1-q}{2-q} \right) \left( \frac{10^{2M}}{a_s^{(2/3)}} \right) \right] \quad ,$$  (3)
where the $q$-value is in the range of $1 < q < 2$, the constant $a_s$ is a proportionality parameter between the
released seismic energy and the fragment size. The $q$-value measures the length scale of spatial





interactions; a $q$-value of ~ 1 indicates short-ranged spatial correlations, and as $q$ increases, the physical
state becomes increasingly unstable. High $q$-values indicate that the fault planes are not in equilibrium,
and more events can be expected (Sotolongo-Costa and Posadas, 2002). In most tectonic regimes, $q$
varies from 1.5 to 1.7 (Sarlis et al., 2010). We calculated the $q$-values of all NVT sequences with
reported magnitudes. The results are shown in Fig. 4 and Table 3.
**3.2.3 Multifractal  detrended moving average analysis**
The multifractal detrended fluctuation analysis (MFDFA) is a technique that quantifies the dynamic
structure of a time series (Kantelhardt et al., 2002). An improvement of the MFDFA was presented by
Gu and Zhou (2010), known as multifractal detrended moving average analysis (MFDMA). We employ
the MFDMA technique to investigate the multifractal properties of interevent time and magnitude
distributions of NVT sequences. We summarize the MFDMA algorithm of Gu and Zhou (2010) as
follows. First, a cumulative time series of a given physical parameter $M(t)$ is constructed and
represented as
$$y(t)=\sum_{i=1}^{t} M(i) \quad ,$$                                              (4)
where $t$ = 1, 2, 3, …, $N$ (length of the time series) and $M(i)$ is the observed time series. Then, a moving
average function of $y(t)$ is calculated in a moving window as
$$\widetilde{y}=\frac{1}{s}\sum_{k=-\lfloor (s-1)\theta\rfloor}^{\lceil (s-1)(1-\theta)\rceil} y(t-k) \quad ,$$                              (5)



where *s* is the window size, $\lfloor x \rfloor$ is the largest integer not greater than argument *x*, $\lceil x \rceil$ is the
smallest integer not smaller than argument *x*, and $\theta$ is the position index which describes the delay
between the moving average function and the original time series ($0 < \theta < 1$). For example, if $\theta = 0, 0.5,$
and 1, it describes a backward, centered, and forward moving average, respectively.
Afterwards, the residual sequences are obtained by detrending the time series through removing the
average function, $\tilde{y}(i)$, resulting in

9        $$\epsilon(i) = y(i) - \tilde{y}(i) \ , \tag{6}$$

where $s \le i \le N$. The residual series ($\epsilon(i)$) is divided into $N_s$ disjoint segments with the same size of
*s*, where $N_s = N/s - 1$. The residual sequence for each segment is denoted by $\epsilon_v$, where

13        $\epsilon_v(i) = \epsilon(l+i)$ for $1 \le i \le s$ and $l = (v-1)s$. Then we calculate the root-mean-square fluctuation

function ($F_v(s)$) for a segment of size *s* as follows

16        $$F_v(s) = \left( \frac{1}{s} \sum_{i=1}^{s} \epsilon_v^2(i) \right)^{1/2} \ . \tag{7}$$

The *q*-th order fluctuation function ($F_q(s)$) is obtained by

20        $$F_q(s) = \left( \frac{1}{N_s} \sum_{v=1}^{N_s} F_v^q(s) \right)^{1/q} \ , \tag{8}$$

for all $q \ne 0$. For the case of $q = 0$, we have



$$\ln\left[F_0(n)\right]=\frac{1}{N_n}\sum_{v=1}^{N_s}\ln\left[F_v(s)\right] \quad , \tag{9}$$

where the scaling behavior of $F_q(s)$ follows the relation that is given by $F_q(s) \sim s^{h(q)}$ where $h(q)$ is the Holder exponent or generalized Hurst exponent. The multifractal scaling exponent is calculated by

$$\tau(q)=q\,h(q)-1 \quad . \tag{10}$$

Finally, the singularity strength function ($\alpha(q)$) and the multifractal spectrum ($f(\alpha)$) can be obtained as

$$\alpha(q)=\frac{d\,\tau(q)}{dq} \quad , \tag{11}$$

and

$$f(\alpha)=q\,\alpha-\tau(q) \quad , \tag{12}$$

respectively. In the multifractal analysis of sequences 1 to 4, we used the following input parameters: $N$ = 30, $\theta = 0$, $q \in \left[-5,5\right]$ with $q$ increments of 0.2, the lower bound of segment size $s$ is fixed to 10, while the upper bound is given by $N/10$, as recommended by Gu and Zhou (2010). In the case of sequences 4 and 5, the upper bound of segment size $s$ is set to $N/4$ because they have less data than the other sequences. Results for $F_q(s)$, $h(q)$, $\tau(q)$, $\alpha(q)$, and $f(\alpha)$ for interevent time and magnitude distributions are shown in Figs. 5 and 6 and Figs. 7 to 9, respectively.





**3.2.4 Multifractal parameters**
We determined multifractal parameters using the equations presented in the previous section, following
De Freitas and França (2024). We start with the degree of asymmetry ($A$), defined as
$$A = \frac{\alpha_{max} - \alpha_0}{\alpha_0 - \alpha_{min}} \quad ,$$ (13)
where $\alpha_0$ is the value for which $f(\alpha)$ is maximum. If $A = 1$, $f(\alpha)$ is symmetric. On the contrary, when A >
1, the symmetry is right-skewed, and if $0 < A < 1$, the symmetry is left-skewed. The values of $\alpha_{max}$ and
$\alpha_{min}$ represent the extreme values of the singularity exponent and are related to the minimum and
maximum fluctuation of the signal. The degree of multifractality ($\Delta\alpha$), which is determined by
$$\Delta\alpha = \alpha_{max} - \alpha_{min} \quad .$$ (14)
A low value of $\Delta\alpha$ denotes that the time series is close to fractal. On the other hand, a high value of $\Delta\alpha$
indicates that the multifractal strength is higher (De Freitas and De Medeiros, 2009). The singularity
parameter ($\Delta f$) describes the broadness of the singularity spectrum and is defined as
$$\Delta f = f(\alpha_{max}) - f(\alpha_{min}) \quad .$$ (15)
In the case that $\Delta f > 1$, the left-hand side is less deep, while if $\Delta f = 0$, both depths of the tails are equal.
According to Ihlen (2012), a long left tail indicates that the singularities are stronger, while in contrast,
a long right tail indicates that the singularities are weaker. The Hurst index ($H$) can be obtained from





the multifractal spectrum through the second-order generalized Hurst exponent $h(q = 2)$. If is $H > 0.5$, it
indicates persistence in long-range correlation, while $H \approx 0.5$ shows a random character of the series
(past and future fluctuations are uncorrelated or Brownian motion). On the other hand, $H < 0.5$ reflects
anti-persistence. In this case, the fluctuations tend not to continue in the same direction, but instead turn
back on themselves, resulting in a less smooth time series (Hampson and Mallen, 2011).
**3.2.5 Sources of multifractality**
Multifractality can be classified into two categories: apparent and intrinsic. The former type refers to
the multifractality that arises from spurious patterns, while the latter refers to the genuine source
derived from nonlinear processes within the data (Saichev and Sornette, 2006). The differentiation
between apparent and intrinsic multifractality is critical for understanding the process present in a time
series (Jiang et al., 2019). On the other hand, there are three primary sources for multifractality in time
series: 1) the non-Gaussian distribution of innovations, 2) the linear long-range correlations, and 3) the
nonlinear long-range correlations (Jiang et al., 2019; Wang, 2023). Two methods are commonly used to
investigate the source of multifractality: shuffling and surrogating procedures, respectively. These
methods involve modifying the original sample to eliminate the source of multifractality. Dealing with
the first source of multifractality is achieved through shuffled time series. The random shuffling of a
time series removes linear and nonlinear temporal correlations (a possible reason for the scaling of $F_q$)
while the probability distribution (PDF) remains unchanged (Kantelhardt, 2009). Thus, if $F_q$ from the
shuffled series scales in the same way as those from the original series, one may assume that the scaling
must be due to the probability distribution of the data. If the shuffled series exhibits weaker
multifractality compared to the original data, then we can assume that multifractality stems from both
temporal correlation and PDF. Consequently, if no multifractal feature remains after performing the
shuffling procedure on the original series, we can interpret that long-range correlation dominates the





multifractality in the original series.
The surrogate time series method generates time series via a Fourier transform, preserving amplitudes
but randomizing the phases, and then performs an inverse Fourier transform. In this form, non-
linearities in the series are eliminated while preserving long-range correlations. The iterated amplitude-
adjusted Fourier Transform (IAAFT) algorithm (Schreiber and Schmitz, 1996; 2000) is appropriate for
this purpose. With the surrogate series, we performed statistical tests for $h(q)$, $\tau(q)$, $\alpha(q)$, and $f(\alpha)$ to
determine the presence or absence of intrinsic multifractality in the data. For this purpose, the tests
consist of determining if the indicator is greater than the indicator derived from the IAAFT series; in
other words, we need to calculate the probability that $x$ is smaller than $x_{IAAFT}$ ($p$-value = $\mathrm{Pr}(x < x_{IAAFT})$),
where x is a multifractal indicator, as proposed by Wang et al. (2023). Wang et al. (2023) also stated
that if the $p$-value is smaller than a significance level (usually 5%), then we can reject the hypothesis
that the original time series is monofractal. The low $p$-values also suggest that the original time series
may have an intrinsic multifractal nature beyond the fat-tailedness and potential long-term linear
correlations. Alternatively, high $p$-values suggest the absence of intrinsic multifractality (De Freitas and
França, 2024). Here, we applied the MFDFA to the magnitude and interevent time series of NVT. In
both cases, we generated 100 shuffled and IAAFT surrogate time series and calculated the multifractal
indicators to analyze the source of multifractality. We used surrogate data to test the presence of
intrinsic multifractality in the time series. The results are presented in Figs. 5 to 9 and Table 4.
**3.2.6 Interevent-time distribution and duration scaling**
Several interevent-time distributions have been proposed in the literature to explain interevent
earthquake observations (e.g., Gamma, Exponential, Lognormal, Weibull) (Corral, 2006; Davidsen and
Kwiatek, 2013). We fitted interevent time data from all the NVT sequences, considering the previously



mentioned statistical distributions, using the maximum likelihood estimation method as described by
Mesimeri et al. (2019). To discern the best goodness of fit, we applied a Kolmogorov-Smirnov test. The
Akaike and Bayesian information criteria (AIC and BIC, respectively) were also calculated to test the
relative quality of the statistical models. The best-fitted distribution is the one with the lowest AIC and
BIC values. The obtained interevent-time probability distributions are shown in Fig. 10 and Table 5.
Additionally, we determined scaling relationships between the tremor duration and magnitude for
sequences 1 to 3 and 5. The obtained scaling relations have the following form:
$\log(\tau) = \alpha M_W + \beta$ ,                                    (16)
where $\tau$ is the duration of the NVT, $M_W$ is the moment magnitude, and $\alpha$ and $\beta$ are constants. We
present the estimated scaling relationships in Fig. 11 and Table 5.
**4 Results**
Our estimates of the $b$-value showed that $b$ for NVTs at the Mexican subduction zone is in the range of
1.25 − 2.42 with completeness magnitudes between 1.10 and 1.80. Sequences 1 and 3 have the highest
$b$-values, indicating a possible individual feature of the down-dip Guerrero segment of the Cocos plate.
They are followed by sequence 2, located at the interface between the Rivera and Cocos plates. The
lowest $b$-value is found for sequence 5 in the Guerrero Gap region (Fig. 2 and Table 2), which
unfortunately can not be corroborated with sequence 6 at the same region since it does not include
magnitude data. The $q$-value from the non-extensive statistical analysis fluctuates from 1.39 to 1.65.
They show an apparent inverse relation with the $b$-value behavior since the sequences located near the
trench (2 and 5) indicate higher $q$-values than those located down-dip (1 and 3) with identical $q$-values.
Multifractal indicators of the original time series show that the multifractal spectra are mostly right-





skewed (magnitude sequences 1 to 4, interevent time sequences 1, 2, and 6). In contrast, left-skewed
spectra correspond to interevent time sequences 3 and 4. Sequence 5 exhibits a symmetric multifractal
spectrum. Results for the degree of multifractality ($\Delta\alpha$) indicate that interevent time series have a
higher multifractal strength than magnitude time series. On the other hand, the singularity parameter
($\Delta f$) indicates that the singularities are weaker for magnitude sequences 2 and 3, as well as interevent
time sequences 1-2 and 6. Contrarily, the singularities are stronger for interevent time sequences 3 and
4. Estimates of the Hurst exponent depict a long-term persistence signature ($H > 0.5$).
By comparing the multifractality spectra ($Fq(s)$, $h(q)$, $\tau(q)$, $\alpha(q)$, and $f(\alpha)$) of shuffled and surrogates
procedures, we observe that they can not destroy the multifractality, indicating that long-term
correlations, the probability distribution of the data, and the presence of nonlinearities are present in the
NVT time sequences. The statistical test results for surrogate data showed that $p$-values for multifractal
indicators ($A$, $\Delta\alpha$, $\Delta f$, and $H$) indicate that magnitude sequence five and interevent time sequences 1, 4,
and 5 exhibit apparent multifractality due to relatively high $p$-values. On the contrary, low $p$-values of
multifractal parameters for magnitude sequence 1 suggest the presence of intrinsic multifractality. In
the cases of magnitude sequences 2 and 3, and interevent time sequences 2, 3, and 6, $p$-values related to
multifractal indicators are inconclusive for determining the nature of multifractality because half of the
indicators exhibit low $p$-values, while the other half exhibit high values. Regarding the inter-event time
distributions, a comparison between the fitted PDFs and the empirical ones from sequences 1 to 4
showed that the Lognormal distribution provides the best fit. In contrast, for sequences 5 and 6, the
Gamma distribution yields the best fit for the interevent time data. In all cases, the least well-fitting
distribution is the Exponential distribution (Fig. 10 and Table 5). Finally, event duration observations
exhibit large scatter, resulting in linear scaling relationships with low determination coefficients ($0.03 <$
$R^2 < 0.34$) (Fig. 11 and Table 6). This scatter is inherent to the genesis of NVT or is associated with the




detection process since it is present in all reported sequences.
**5 Discussion**
We start the discussion by comparing our $b$-value estimates with previous studies. The $b$-value
associated with NVTs has been determined in crustal and subduction environments, with the latter
being the most common. In crustal regions, for example, in the San Andreas Fault at the Parkfield
segment, Staudenmaier et al. (2019) calculated a $b$-value of 2.52 for NVT episodes. For the case of
subduction zones, it has been found that the $b$-value of NVTs fluctuates between 1.0 and 5 (Kao et al.,
2010; Rabbel et al., 2011; Gallego et al., 2013; Sweet et al., 2014; Bostock et al., 2015). The Cascadia
subduction zone, in Vancouver Island, exhibits both low and high $b$-values ($b \sim 1$ and $4.2 < b < 5$,
respectively) (Kao et al., 2010; Sweet et al., 2014; Bostock et al., 2015). In Chile, the $b$-value of NVT
sequences was determined as 2.4 (Gallego et al., 2013). Regarding $b$-value estimations of NVTs at the
Cocos plate, Rabbel et al. (2011) estimated a value of 1 in the region of Costa Rica. Our results show
that the $b$-value varies from 1.25 to 2.42 for the NVTs detected at the Mexican subduction regimes
studied. We observed that in the down-dip segment, NVT sequences have the highest $b$-values (2.22-
2.42), while in the inter-plate coupling region, the $b$-value is in the range of $1.25 - 1.41$ (Table 2). The
high $b$-values can be evidence of a larger degree of fracturing since they convey a larger proportion of
small magnitude fractures as compared to larger ones. In general, the $b$-values obtained for the coupling
region do not depart much from common tectonic events' $b$-value estimations. For example, for the
Cocos plate, the $b$-value for tectonic seismic events has been observed to lie in the interval of $0.8 - 1.3$
(Nishikawa and Ide, 2014). In terms of the non-extensive statistical analysis, our estimates of the $q$-
value for NVT in Mexico are consistent with reports of $q$-value at subduction zones in which values of
$q$ fluctuate from 1.61 to 1.69 (Scherrer et al., 2015). These reports agree with our results, in which the
$q$-value varies from 1.64 to 1.65. Our results also showed that interplate down-dip sequences have




lower *q*-values (1.39) (Table 3). High *q*-values of coastal regions can be explained by stress
heterogeneity due to plate coupling and asperity distribution compared to interplate down-dip regions
with different conditions, such as pressure, temperature, and rock structure.
Regarding the multifractal analysis, our results indicate that both magnitude and interevent time NVT
sequences of NVT exhibit a multifractal structure, in a similar form to tectonic earthquakes (interevent
time, Michas et al., 2015; magnitude, De Freitas and França, 2024). In relation to the identification of
intrinsic multifractality, the tests of surrogate data verify that only one sequence exhibits intrinsic
multifractality. On the contrary, four sequences confirm the presence of apparent multifractality, while
in five sequences, the results are not conclusive. De Freitas and França (2024) also reported that the
seismicity of some subduction zones has apparent multifractality, while others exhibit intrinsic
multifractality. These observations agree with our findings. Conversely, our estimates of the Hurst
exponent showed that *H* varies from 0.65 to 1.06. Most of the NVT sequences exhibit an exceptionally
high persistent memory (*H* > 0.95) (Table 4). According to Cisternas et al. (2004), high *H* values may
be associated with the volumes around the ruptured faults. Here, we interpreted high Hurst exponents
as the result of the relatively limited volume of perturbed regions where the fluids are present. Our
results are also in agreement with Hurst exponent reports for regional seismicity and aftershock
sequences. For example, in southern Italy, Telesca et al. (2001) determined that *H* fluctuates from 0.5 to
0.92; in Taiwan and Greece, *H* is about 0.8 (Chen et al., 2008; Gkarlaouni et al., 2017, respectively),
while in the San Andes fault zone in California, *H* is equal to 0.87 (De Freitas et al., 2013). In the case
of the aftershocks of the 1999 Izmit earthquake ($M_W$ = 7.6), the Hurst exponent has a value of 0.95
(Cisternas et al., 2004).
The interevent time analysis showed that NVT sequences 1 to 4 are well approximated by a Lognormal



distribution, while for sequences 5 and 6, by a Gamma distribution (Table 5). This shows that the inter-
event time distributions exhibit a mixed behavior, with characteristics compatible with tectonic
earthquakes and volcanic seismicity. For example, Traversa and Grasso (2010) showed that a Gamma
distribution can mainly describe volcano seismicity, but some episodes reject the Gamma distribution
to describe the seismic activity. In the case of the tectonic activity, several inter-event time density
distributions have been proposed, such as the Exponential, Gamma, Lognormal, etc (Corral, 2006;
Passarelli et al., 2015; Post et al., 2021). Our results are also consistent with global ($M_\mathrm{W} \geq 7$, Bantidi,
2022) and regional ($0.1 < M_\mathrm{L} < 5.1$, Mesimeri et al., 2019) seismicity studies, in which Lognormal fits
best the interevent time observations. Additionally, our results depict a differentiated behavior between
the down-dip and the interplate coupling regions. In the last case, the Gamma distribution is
highlighted as explaining the observations better, possibly due to the smaller sample size (Table 1).
Finally, the obtained event duration scaling relationships for all the NVT sequences exhibit large
scatter. The data variability may be inherent to the genesis of NVT or be associated with the detection
process since it is present in all reported sequences. It is a fact that tremors do not have clear phase
arrivals, resulting in longer durations. On the other hand, determining reliable tremor parameters such
as location, magnitude, peak amplitudes, and duration is challenging compared to regular earthquakes
(Staudenmaier et al., 2019). The studied data highlight that the duration of the events varies from
tremor episodes, even in the same subduction zone. According to Schwartz and Rokosky (2007), the
mechanism that controls the duration and amplitude of tremors is not clearly known, but it is suggested
that the presence of fluids may explain the tremor source. Although the relationship between slow
earthquakes and tectonic tremors is not known with certainty, there is evidence that NVTs are
modulated for slow dislocations on the plate interface (Villafuerte and Cruz-Atienza, 2017).



**6 Conclusions**

We studied statistical features of reported NVT sequences along the Mexican subduction zone. We analyzed the Gutenberg-Richter relation, non-extensive statistics, and multifractal properties of magnitude and interevent time series. We observed that $b$-values are in the range of $1.25 - 2.42$, being consistent with reports worldwide. In particular, down-dip region NVT sequences have the highest $b$-values. In contrast, $q$-values of down-dip regions have lower values than those from coastal areas ($1.39 < q < 1.65$). Both magnitude and interevent time sequences exhibit multifractality, similarly to tectonic earthquakes, that may be related to long-term correlations, the probability distribution of the data, and the presence of nonlinearities. By analyzing surrogate data, we observed that some sequences exhibit apparent multifractality, and to a lesser extent intrinsic multifractality, although in other cases the criteria were not conclusive. Regarding the study of the PDFs, we find that most sequences can be described by a Lognormal distribution and in two cases by a Gamma distribution. This suggests that NVT sequences exhibit a mixed behavior with characteristics similar to tectonic earthquakes and volcanic seismicity. Concerning the NVT duration observations, a large scatter is observed, resulting in linear scaling relations with low determination coefficients. The variability in NVT durations may reflect an inherent origin or may be associated with the difficulties in the detection process and/or waveform parameter estimations.

*Code availability*. Estimations of $b$-value were performed with the Python code Calc_gr_ks (https://github.com/nadavwetzler/b-value; Wetzler, 2022). We calculate the interevent-time distributions with the code qks_statistics (https://github.com/mmesim/qks_statistics; Mesimeri, 2021). The MFDMA method was implemented with the code GFGU_MFDMA_1D (Gu and Zhou, 2010). MATLAB Chaotic Systems Toolbox for implementing shuffle and IAAFT surrogate time series are available at https://github.com/nmitrou/Simulations/tree/master/matlab_codes (Leontitsis, 2001).



Linear regressions for NVT duration scaling relations were determined with the SciPy library (Virtanen
et al., 2020). Some figures were produced with Generic Mapping Tools (GMT) (Wessel et al., 2019). In
all cases, last access: September 2025.
*Data availability*. NVT catalogs for sequences 1 to 3 were taken from the World Tremor Database
(http://www-solid.eps.s.u-tokyo.ac.jp/~idehara/wtd0/Welcome.html, Ide, 2012; Idehara et al., 2014).
Catalogs for sequences 4, 5, and 6 were obtained from Husker et al. (2019), Plata-Martínez et al.
(2021), and  Chen et al. (2025), respectively. In all cases, last access: September 2025.
*Author contributions*. Conceptualization: QRP. Data analysis: QRP, VHMR, FRZ. Writing and
discussion of the manuscript: QRP, VHMR, FRZ.
*Competing interests*. The contact author has declared that none of the authors has any competing
interests.
*Financial support*. QRP was supported by the Secretaría de Ciencia, Humanidades, Tecnología e
Innovación (SECIHTI) (project 7197).

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



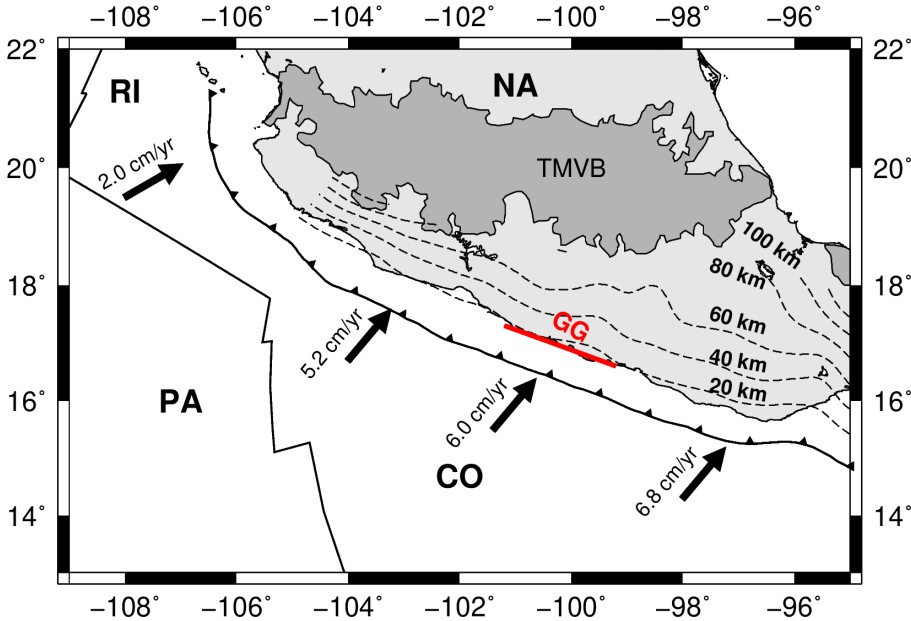

**Figure 1.** Tectonic framework of the Mexican subduction zone. TMVB is the Trans-Mexican Volcanic

Belt. CO, NA, PA, and RI are the Cocos, North American, Pacific, and Rivera plates, respectively. GG

is the Guerrero seismic gap. Black arrows indicate the convergence rate among tectonic plates. Dashed

lines represent contour lines of the subducted plate, spaced every 20 km, from 20 to 100 km (Hayes,

2012).



**Figure 2.** Hypocenter locations of all the studied non-volcanic tremor sequences along the Mexican subduction zone. (**a**) sequence 1, (**b**) sequence 2, (**c**) sequence 3, (**d**) sequence 4, (**e**) sequence 5, and (**f**) sequence 6.

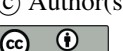

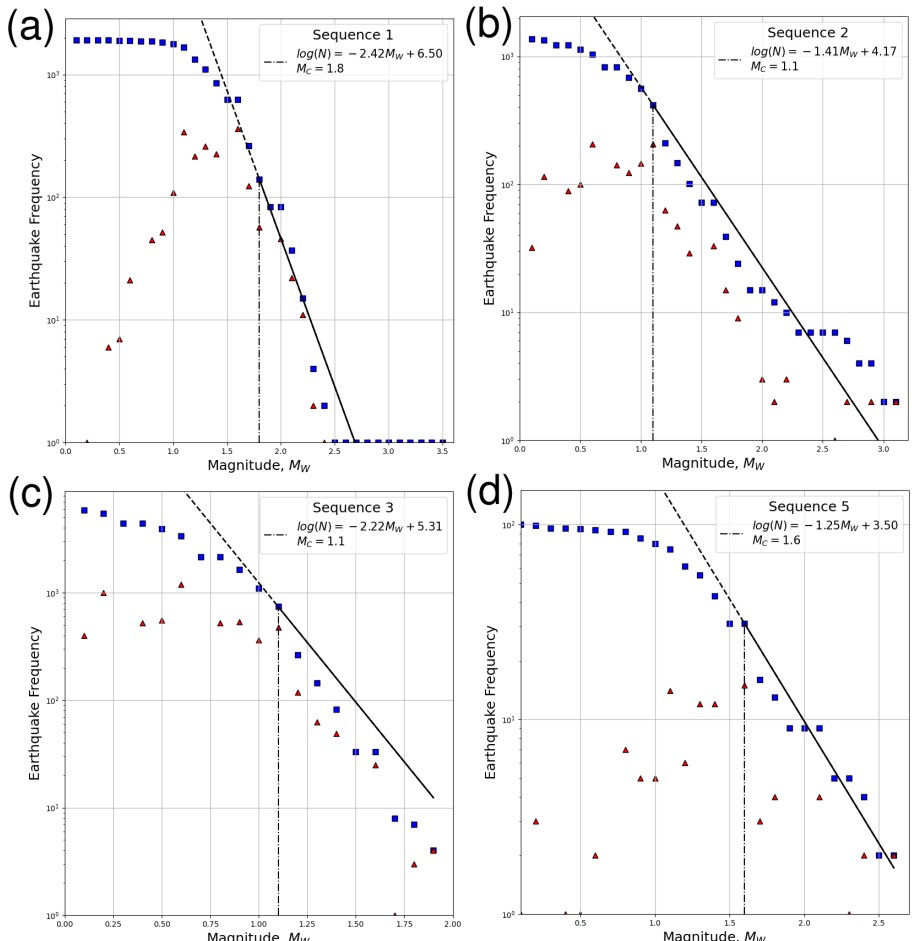

**Figure 3.** Estimates of *b*-value and $M_c$ for the studied NVT sequences. Blue squares show the cumulative number of events versus magnitude. Red triangles exhibit the number of events. The solid black lines indicate the Gutenberg-Righter frequency magnitude distributions. (**a**) sequence 1, (**b**) sequence 2, (**c**) sequence 3, and (**d**) sequence 5.



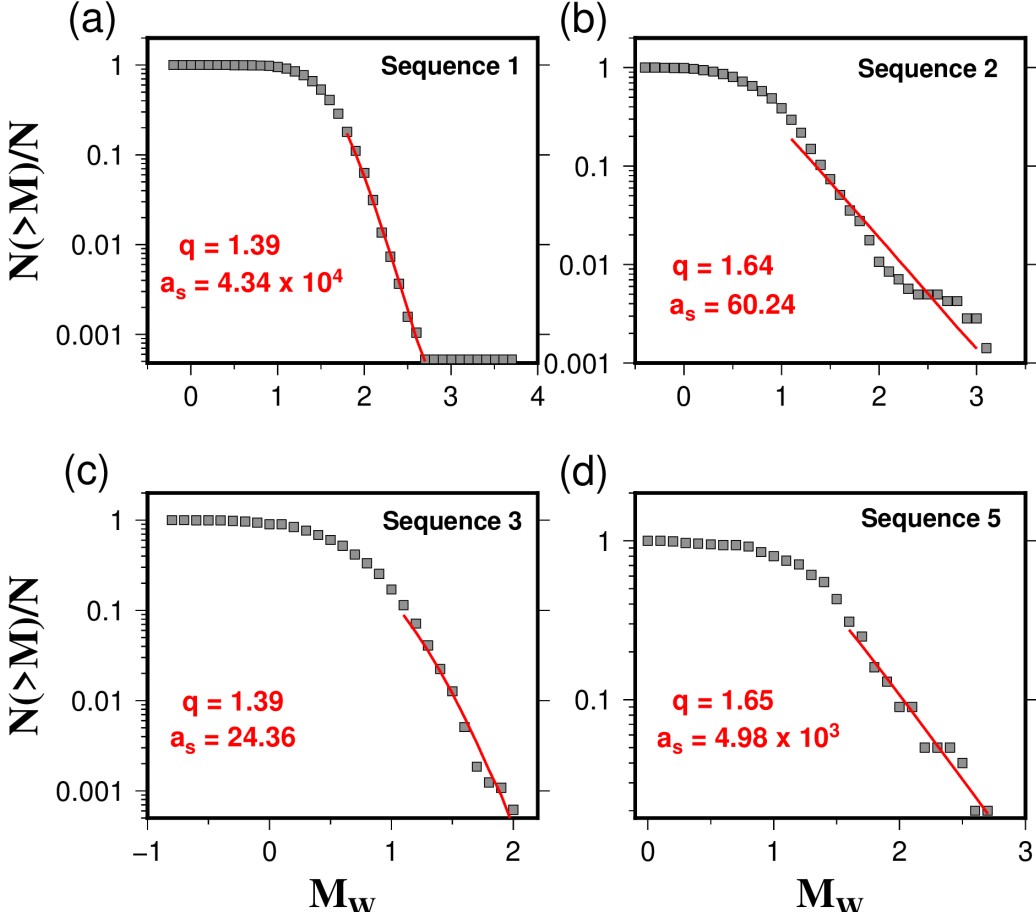

2 **Figure 4.** Normalized cumulative number of earthquake with magnitude $m > M_{th}$ and fitting obtained

3 with the fragment asperity model for the seismicity monitored in each NVT sequence. (**a**) sequence 1,

4 (**b**) sequence 2, (**c**) sequence 3, and (**d**) sequence 5.





**Figure 5.** Multifractal analysis of magnitude for sequences 1 (**a**) and 2 (**b**) (fluctuation function, $F(q)$;

Hurst exponent, $h(q)$; multifractal scaling exponent $\tau(q)$; and multifractal spectrum $f(\alpha)$). In all cases,

the original, shuffled, and IAAFT surrogates data are shown in red, green, and blue color, respectively.







**Figure 6.** Multifractal analysis of magnitude for sequences 3 (**a**) and 5 (**b**) (fluctuation function, $F(q)$; Hurst exponent, $h(q)$; multifractal scaling exponent $\tau(q)$; and multifractal spectrum $f(\alpha)$). In all cases, the original, shuffled, and IAAFT surrogates data are shown in red, green, and blue color, respectively.







**Figure 7.** Multifractal analysis of interevent time for sequences 1 (**a**) and 2 (**b**) (fluctuation function,

$F(q)$; Hurst exponent, $h(q)$; multifractal scaling exponent $\tau(q)$; and multifractal spectrum $f(\alpha)$). In all

cases, the original, shuffled, and IAAFT surrogates data are shown in red, green, and blue color,

respectively.







**Figure 8.** Multifractal analysis of interevent time for sequences 3 (**a**) and 4 (**b**) (fluctuation function,

$F(q)$; Hurst exponent, $h(q)$; multifractal scaling exponent $\tau(q)$; and multifractal spectrum $f(\alpha)$). In all

cases, the original, shuffled, and IAAFT surrogates data are shown in red, green, and blue color,

respectively.





**Figure 9.** Multifractal analysis of interevent time for sequences 5 (**a**) and 6 (**b**) (fluctuation function,
$F(q)$; Hurst exponent, $h(q)$; multifractal scaling exponent $\tau(q)$; and multifractal spectrum $f(\alpha)$). In all
cases, the original, shuffled, and IAAFT surrogates data are shown in red, green, and blue color,
respectively.





**Figure 10.** Probability density functions of the interevent times for the NVT sequences and fitted curves of different statistical distributions (exponential, Gamma, Lognormal, and Weibull). (**a**) sequence 1, (**b**) sequence 2, (**c**) sequence 3, (**d**) sequence 4, (**e**) sequence 5, and (**f**) sequence 6.





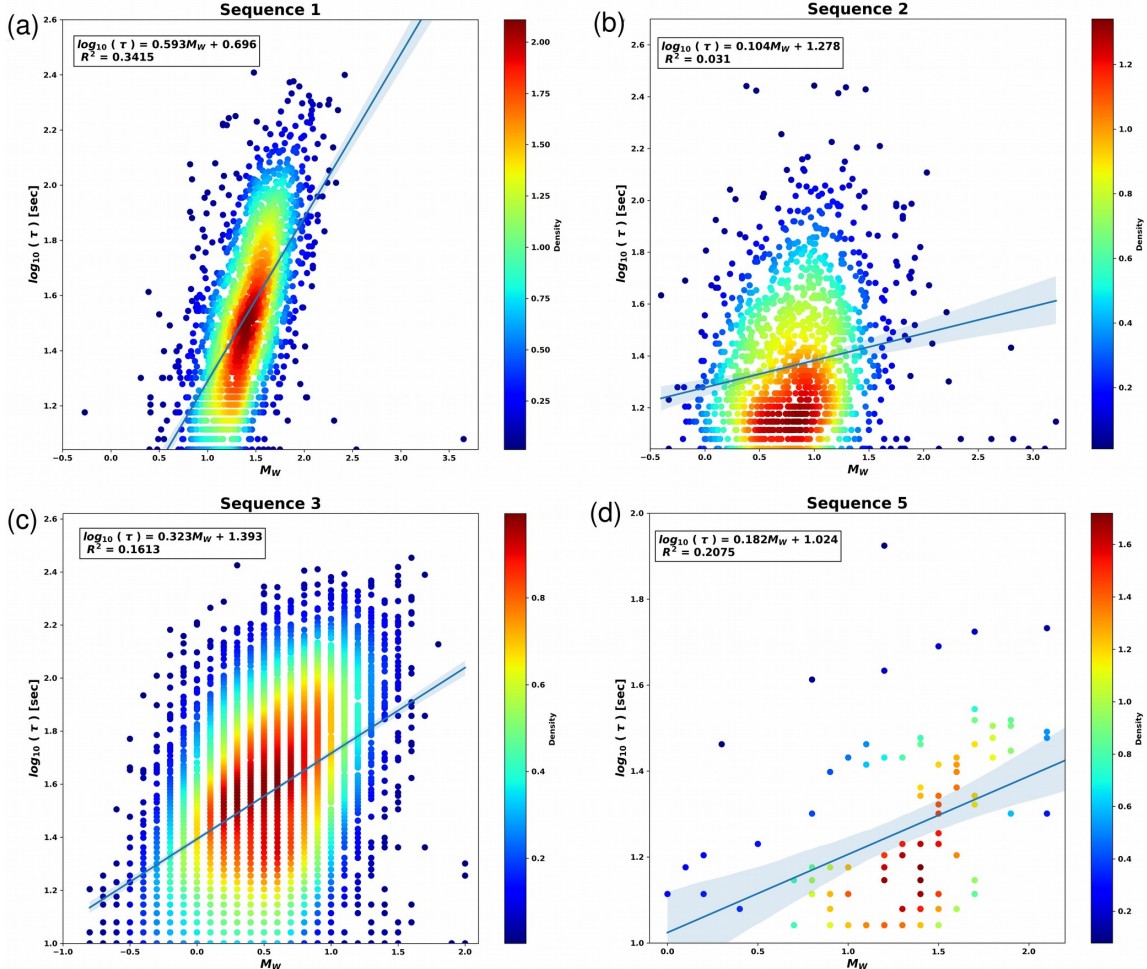

**Figure 11.** NVT duration scaling relationships (blue lines). In all figures, color indicates the density of

NVT observations. (**a**) sequence 1, (**b**) sequence 2, (**c**) sequence 3, and (**d**) sequence 5.



**Table 1.** Studied non-volcanic tremor sequences. $N$ is the number of events; $M_W$ is the moment
magnitude; $T_1$ and $T_2$ are the start and end dates of the located NVTs.

| Sequence | N | $T_1$ yyyy/mm/dd | $T_2$ yyyy/mm/dd | $M_W$ | Reference |
|---|---|---|---|---|---|
| 1 | 1908 | 2005/01/14 | 2007/05/28 | -0.27 – 3.65 | Ide (2012), Idehara et al. (2014) |
| 2 | 1411 | 2006/01/26 | 2007/06/06 | -0.40 – 3.21 | Ide (2012), Idehara et al. (2014) |
| 3 | 6776 | 2009/11/26 | 2013/08/11 | -0.80 – 2.00 | Ide (2012), Idehara et al. (2014) |
| 4 | 23408 | 2009/03/04 | 2022/05/29 | | Husker et al. (2019) |
| 5 | 101 | 2017/11/22 | 2018/11/12 | 0.10 – 2.70 | Plata-Martínez et al. (2021) |
| 6 | 637 | 2017/11/21 | 2019/09/10 | | Chen et al. (2025) |

**Table 2.** Results for the Gutenberg-Richter relationship.

| Sequence | $b$-value | $a$-value | $M_c$ |
|---|---|---|---|
| 1 | 2.42 | 6.50 | 1.80 |
| 2 | 1.41 | 4.17 | 1.10 |
| 3 | 2.22 | 5.31 | 1.10 |
| 5 | 1.25 | 3.50 | 1.60 |

**Table 3.** Results for the fragment asperity model.

| Sequence | $q$-value | $a_s$-value |
|---|---|---|
| 1 | 1.39 | $4.34 \times 10^4$ |
| 2 | 1.64 | 60.24 |
| 3 | 1.39 | 24.36 |
| 5 | 1.65 | $4.98 \times 10^3$ |

**Table 4.** Testing multifractality of multifractal parameters. $M_W$ is the moment magnitude; $\Delta t$ is the
interevent times; $A$ is the degree of asymmetry; $\Delta\alpha$ is the degree of multifractality; $\Delta f$ is the singularity
parameter; and $H$ is the Hurst index. The symbol $\langle\rangle$ denotes the mean and σ indicates the standard
deviation. The $p$-values represent the proportion of IAAFT surrogates measured for each indicator that
exceeds the index's value for the original time series.

| Parameter | Sequence | $A$ | $\langle A \rangle$ | $\sigma_A$ | $p$-value | $\Delta\alpha$ | $\langle \Delta\alpha \rangle$ | $\sigma_{\Delta\alpha}$ | $p$-value | $\Delta f$ | $\langle \Delta f \rangle$ | $\sigma_{\Delta f}$ | $p$-value | $H$ | $\langle H \rangle$ | $\sigma_H$ | $p$-value |
|---|---|---|---|---|---|---|---|---|---|---|---|---|---|---|---|---|---|
| $M_W$ | 1 | 2.96 | 1.16 | 0.14 | 0.00 | 0.04 | 0.03 | 0.002 | 0.00 | -0.060 | -0.006 | 0.005 | 0.00 | 1.03 | 1.03 | 0.002 | 0.47 |
| | 2 | 2.83 | 2.31 | 1.06 | 0.25 | 0.26 | 0.23 | 0.070 | 0.25 | -0.370 | -0.254 | 0.203 | 0.77 | 1.02 | 1.02 | 0.006 | 0.58 |
| | 3 | 4.67 | 3.91 | 0.43 | 0.05 | 0.80 | 0.69 | 0.061 | 0.05 | -0.980 | -0.822 | 0.118 | 0.91 | 0.98 | 0.99 | 0.005 | 0.94 |
| | 5 | 1.04 | 1.11 | 0.29 | 0.56 | 0.04 | 0.06 | 0.014 | 0.83 | -0.002 | -0.007 | 0.022 | 0.43 | 1.06 | 1.06 | 0.012 | 0.39 |
| $\Delta t$ | 1 | 1.82 | 1.73 | 0.36 | 0.42 | 1.71 | 1.70 | 0.204 | 0.50 | -0.780 | -0.452 | 0.311 | 0.85 | 0.82 | 0.86 | 0.016 | 0.99 |
| | 2 | 3.16 | 2.48 | 0.66 | 0.14 | 1.73 | 1.32 | 0.194 | 0.03 | -0.810 | -0.724 | 0.289 | 0.62 | 0.92 | 0.93 | 0.015 | 0.80 |
| | 3 | 0.65 | 0.62 | 0.13 | 0.43 | 1.60 | 1.23 | 0.128 | 0.00 | 0.413 | 0.312 | 0.203 | 0.34 | 0.65 | 0.83 | 0.025 | 0.00 |
| | 4 | 0.70 | 0.88 | 0.09 | 0.96 | 0.80 | 0.89 | 0.050 | 0.98 | 0.125 | 0.129 | 0.132 | 0.49 | 0.86 | 0.90 | 0.004 | 0.00 |
| | 5 | 1.02 | 1.32 | 0.79 | 0.58 | 0.72 | 0.84 | 0.256 | 0.65 | 0.031 | -0.122 | 0.490 | 0.41 | 0.96 | 0.97 | 0.060 | 0.61 |
| | 6 | 3.90 | 1.49 | 0.58 | 0.00 | 0.90 | 0.50 | 0.090 | 0.00 | -1.044 | -0.210 | 0.224 | 0.99 | 0.96 | 0.99 | 0.021 | 0.86 |



**Table 5.** Estimated parameters for the PDF of interevent times. AIC and BIC are the Akaike and Bayesian information criteria; $D$ is the test statistic.

| Sequence | Distribution | Parameters | -logL | K-S test p-value | D | AIC | BIC |
|---|---|---|---|---|---|---|---|
| 1 | Lognormal | $\mu = -3.65$ $\sigma = 2.27$ | -2696 | 0.70 | 0.10 | -5389 | -5385 |
| | Weibull | $\alpha = 0.09$ $b = 0.40$ | -2399 | 0.31 | 0.14 | -4795 | -4791 |
| | Gamma | $\alpha = 0.25$ $b = 1.78$ | -2020 | 0.02 | 0.23 | -4036 | -4033 |
| | Exponential | $\mu = 0.44$ | 359 | $2.50\times10^{-13}$ | 0.58 | 719 | 721 |
| 2 | Lognormal | $\mu = -3.26$ $\sigma = 2.30$ | -1424 | 0.83 | 0.09 | -2845 | -2841 |
| | Weibull | $\alpha = 0.12$ $b = 0.44$ | -1308 | 0.39 | 0.14 | -2611 | -2608 |
| | Gamma | $\alpha = 0.31$ $b = 1.11$ | -1191 | 0.08 | 0.20 | -2378 | -2375 |
| | Exponential | $\mu = 0.35$ | 84 | $3.81\times10^{-9}$ | 0.48 | -166 | -164 |
| 3 | Lognormal | $\mu = -4.34$ $\sigma = 1.88$ | -15527 | 0.41 | 0.13 | -31051 | -31047 |
| | Weibull | $\alpha = 0.04$ $b = 0.44$ | -14009 | 0.09 | 0.18 | -28014 | -28011 |
| | Gamma | $\alpha = 0.26$ $b = 0.76$ | -12035 | 0.0009 | 0.28 | -24066 | -24062 |
| | Exponential | $\mu = 0.20$ | -4202 | $2.58\times10^{-16}$ | 0.61 | -8402 | -8400 |
| 4 | Lognormal | $\mu = -3.30$ $\sigma = 1.64$ | -32533 | 0.62 | 0.12 | -65063 | -65059 |
| | Weibull | $\alpha = 0.09$ $b = 0.54$ | -28883 | 0.18 | 0.17 | -57762 | -57759 |
| | Gamma | $\alpha = 0.38$ $b = 0.53$ | -24665 | 0.0064 | 0.27 | -49326 | -49323 |
| | Exponential | $\mu = 0.20$ | -13681 | $2.09\times10^{-8}$ | 0.48 | -27359 | -27358 |
| 5 | Lognormal | $\mu = -0.39$ $\sigma = 2.86$ | 208 | 0.07 | 0.22 | 419 | 423 |
| | Weibull | $\alpha = 2.25$ $b = 0.53$ | 188 | 0.47 | 0.14 | 382 | 384 |
| | Gamma | $\alpha = 0.39$ $b = 9.30$ | 185 | 0.68 | 0.12 | 374 | 377 |
| | Exponential | $\mu = 3.65$ | 229 | 0.05 | 0.23 | 461 | 462 |
| 6 | Lognormal | $\mu = -2.35$ $\sigma = 3.74$ | 246 | 0.07 | 0.21 | 495 | 499 |
| | Weibull | $\alpha = 0.49$ $b = 0.40$ | 139 | 0.26 | 0.16 | 283 | 286 |
| | Gamma | $\alpha = 0.29$ $b = 3.54$ | 76 | 0.42 | 0.14 | 156 | 159 |
| | Exponential | $\mu = 1.03$ | 655 | 0.01 | 0.25 | 1312 | 1314 |



1  **Table 6.** Duration scaling relationships ($\log(\tau) = a + b\, M_{\mathrm{w}}$). $R^2$ is the determination coefficient; $b$ is the
2  slope and $a$ is the intercept.

| Sequence | $a$ | $b$ | $R^2$ |
|---|---|---|---|
| 1 | 0.696 | 0.593 | 0.341 |
| 2 | 1.278 | 0.104 | 0.031 |
| 3 | 1.393 | 0.323 | 0.161 |
| 5 | 1.024 | 0.182 | 0.208 |

