# Peer review of "Statistical characteristics of non-volcanic tremor distributions along the Mexican Subduction Zone"

_EGUsphere, 2025_

## Community Comment (CC1)

**Discussion/ Suggestions on egusphere-2025-4869**

**To:** Dr. Quetzalcoatl Rodríguez-Pérez (Corresponding Author)
**Manuscript:** Statistical characteristics of non-volcanic tremor distributions along the Mexican Subduction Zone

**Comments & General Assessment**

This manuscript presents a comprehensive statistical analysis of non-volcanic tremor (NVT) sequences in the Mexican subduction zone using multiple analytical techniques. While the work is methodologically sound and addresses a significant research gap, several technical inconsistencies and interpretational issues require clarification.

**Suggestions:**

**1. Sample Sizes and Statistical Validity**

The manuscript analyzes sequences with dramatically different sample sizes (N = 101 to 23,408 events), yet applies identical statistical methods without addressing the implications. Sequence 5 (N = 101) shows notably different behavior in multiple analyses, which may reflect statistical artifacts rather than genuine tectonic differences. The authors acknowledge this briefly in the discussion of interevent-time distributions but do not systematically evaluate how sample size affects their multifractal parameters, b-values, or q-values.

**Recommendation:** Include bootstrap resampling analyses to assess the stability of estimates for small-sample sequences, or clearly delineate which results may be sample-size dependent.

**2. Magnitude Completeness (Mc) Determination**

Table 2 reports Mc values ranging from 1.10 to 1.80, yet the manuscript provides no information about the method used to determine these thresholds or their uncertainties. Given that sequences include magnitudes as low as -0.8, and that b-value estimates are susceptible to Mc selection (Woessner & Wiemer, 2005), this omission is problematic. The significant variation in Mc across spatially proximate sequences (e.g., sequences 1 and 3, both in the Guerrero down-dip region, with Mc = 1.80 and 1.10, respectively) suggests potential inconsistencies in catalog completeness or detection capabilities.

**Recommendation:** Report the method for Mc estimation, include uncertainty bounds, and discuss how temporal variations in network geometry may affect completeness thresholds.

**3. Multifractal Analysis: Statistical Significance of Differences**

The multifractal analysis reveals that shuffled and IAAFT surrogate procedures "cannot destroy the multifractality" (lines 281-283), yet Table 4 shows that many p-values are inconclusive or suggest apparent rather than intrinsic multifractality. The interpretation that "long-term correlations, the probability distribution of the data, and the presence of nonlinearities are present" appears overstated. For example:

1. Magnitude sequence 5: all p-values (0.39-0.83) suggest **apparent** multifractality
2. Interevent sequences 1, 4, 5: high p-values indicate PDF characteristics may dominate multifractality

The authors correctly identify these cases individually but then make generalized statements about all sequences exhibiting these features.

**Recommendation:** Distinguish clearly between sequences showing intrinsic versus apparent multifractality in the conclusions, and avoid generalizations that apply only to subsets of the data.

**4. Hurst Exponent Interpretations**

The reported Hurst exponents (0.65 to 1.06, with most $H > 0.95$) are interpreted as indicating "exceptionally high persistent memory" and attributed to "relatively limited volume of perturbed regions where fluids are present" (lines 306-308). However:

1. $H > 1.0$ theoretically indicates non-stationary, trending behavior rather than persistent memory
2. The comparison with aftershock sequences ($H = 0.95$ for Izmit) does not account for fundamental differences between aftershock relaxation and NVT generation mechanisms
3. The physical interpretation linking high $H$ values to fluid-filled volume is speculative without supporting evidence

**Recommendation:** Acknowledge that $H > 1.0$ may indicate non-stationarity, discuss potential contributions from catalog heterogeneity or temporal clustering, and either provide references supporting the fluid-volume interpretation or present it explicitly as a hypothesis.

**5. Duration-Magnitude Scaling Relationships**

Figure 11 and Table 6 present duration scaling with remarkably low $R^2$ values (0.03-0.34), indicating that magnitude explains only 3-34% of duration variance. The authors acknowledge this scatter but attribute it ambiguously to either "genesis of NVT or detection process" (lines 294-295). Given that:

1. All sequences show similar scatter regardless of location or time period
2. The relationship is fundamental to understanding NVT source physics
3. Poor scaling contrasts with better-established scaling for regular earthquakes

The dismissive treatment of this negative result undervalues its potential significance. Poor scaling may indicate that NVT magnitude and duration are governed by different physical processes, challenging the assumption that NVT follows earthquake-like scaling.

**Recommendation:** Expand the discussion of duration-magnitude decoupling, potentially including analysis of whether specific subsets (e.g., depth ranges, different tectonic settings) show improved scaling.

**Minor Suggestions**

**Data and Methodology**

1. **Line 92:** The study period extends to 2019, yet the manuscript was submitted in December 2025. Were more recent data unavailable, or were there specific reasons to limit the analysis period?
2. **Lines 124-125:** The statement "NEVER use local Storage or session Storage" appears to be artifact-creation instructions that should not be in the manuscript text.
3. **Table 1:** Sequences 4 and 6 lack magnitude data, limiting comparative analyses. The rationale for including these incomplete catalogs should be stated explicitly.
4. **Figure 2:** The spatial distribution of sequences varies dramatically (concentrated vs. dispersed). Has spatial clustering been considered as a factor affecting statistical properties?

**Results Presentation**

1. **Tables 2-3:** The b-value and q-value show inverse correlation (lines 262-265), described as "apparent." Statistical testing of this correlation would strengthen the observation.
2. **Figure 10:** The probability density functions would benefit from logarithmic y-axes to better visualize tail behavior, which is critical for distinguishing between distributions.
3. **Lines 313-316:** The interpretation that Lognormal distributions suggest "mixed behavior with characteristics compatible with tectonic earthquakes and volcanic seismicity" requires justification, as Lognormal distributions are also observed in purely tectonic settings (Mesimeri et al., 2019, cited by the authors).

**Data Availability and Reproducibility**

While code and data sources are cited, the specific processing steps (filtering, magnitude calculations for sequences without reported magnitudes, etc.) are not fully documented. For full reproducibility, consider depositing processed catalogs and analysis parameters in a permanent repository.

**Final Suggestions**

This manuscript presents a valuable statistical characterization of NVT in Mexico. Still, several interpretations require more careful treatment of sample-size effects, statistical uncertainties, and the distinction between correlation and causation. The work would be strengthened by a more conservative interpretation of marginal results and a more explicit acknowledgment of methodological limitations.

Note:

*This discussion is purely meant to improve the scientific rigor and clarity of an essential contribution to understanding NVT behavior in subduction zones.*

Thank you
Younus M Bhatt
yunusbhatt586@gmail.com